# The Wasserstein Proximal Gradient Algorithm

**Adil Salim**
Visual Computing Center
KAUST
adil.salim@kaust.edu.sa

**Anna Korba**
Gatsby Computational Neuroscience Unit
University College London
a.korba@ucl.ac.uk

**Giulia Luise**
Computer Science Department
University College London
g.luise16@ucl.ac.uk

## Abstract

Wasserstein gradient flows are continuous time dynamics that define curves of steepest descent to minimize an objective function over the space of probability measures (*i.e.*, the Wasserstein space). This objective is typically a divergence w.r.t. a fixed target distribution. In recent years, these continuous time dynamics have been used to study the convergence of machine learning algorithms aiming at approximating a probability distribution. However, the discrete-time behavior of these algorithms might differ from the continuous time dynamics. Besides, although discretized gradient flows have been proposed in the literature, little is known about their minimization power. In this work, we propose a Forward Backward (FB) discretization scheme that can tackle the case where the objective function is the sum of a smooth and a nonsmooth geodesically convex terms. Using techniques from convex optimization and optimal transport, we analyze the FB scheme as a minimization algorithm on the Wasserstein space. More precisely, we show under mild assumptions that the FB scheme has convergence guarantees similar to the proximal gradient algorithm in Euclidean spaces.

## 1 Introduction

The task of transporting an initial distribution $\mu_0$ to a target distribution $\mu_\star$ is common in machine learning. This task can be reformulated as the minimization of a cost functional defined over the set of probability distributions. Wasserstein gradient flows [2] are suitable continuous time dynamics to minimize such cost functionals. These flows have found applications in various fields of machine learning such as reinforcement learning [32, 40], sampling [5, 12, 16, 38] and neural networks optimization [14, 26]. Indeed, Wasserstein gradient flows can be seen as the continuous limit of several discrete time machine learning algorithms. The analysis of continuous time dynamics is often easier than the analysis of their discrete time counterparts. Therefore, many works focus solely on continuous time analyses of machine learning algorithms such as variants of gradient descent [7, 14, 15, 26, 34, 37]. Besides, although discretized Wasserstein gradient flows have been proposed in the literature [2, 8, 11, 19, 24, 38], most of them have not been studied as minimization algorithms.

In this paper, we focus on the resolution, by a discrete time algorithm, of a minimization problem defined on the set $\mathcal{P}_2(\mathcal{X})$ of probability measures $\mu$ over $\mathcal{X} = \mathbb{R}^d$ such that $\int \|x\|^2 d\mu(x) < \infty$. More precisely, $\mu_\star$ is defined as a solution to

$$\min_{\mu \in \mathcal{P}_2(\mathcal{X})} \mathcal{G}(\mu) := \mathcal{E}_F(\mu) + \mathcal{H}(\mu), \tag{1}$$

where $\mathcal{E}_F$ is a potential energy $\mathcal{E}_F(\mu) = \int F(x)d\mu(x)$ tied to a smooth convex function $F : \mathbb{R}^d \to \mathbb{R}$, and $\mathcal{H}$ is a nonsmooth term *convex along the generalized geodesics* defined by the Wasserstein distance. The potential $\mathcal{E}_F$ plays the role of a data fitting term whereas $\mathcal{H}$ can be seen as a regularizer.

**Motivation for studying the template problem** (1). Many free energy minimization problems can be cast as Problem (1), see [1, Proposition 7.7] or more generally [2, Section 9]. For instance, $\mathcal{H}$ can be an internal energy [2, Example 9.3.6]. In particular, if $\mathcal{H}$ is the negative entropy, then $\mathcal{G}$ boils down to the Kullback Leibler divergence (up to an additive constant) w.r.t. the Gibbs measure $\mu_\star \propto \exp(-F)$. This remark has been used by several authors to study sampling tasks as minimizing Problem (1) [5, 12, 16, 20, 38]. Another example where $\mathcal{H}$ is an internal energy is the case where $\mathcal{H}$ is a higher order entropy. In this case, $\mu_\star$ follows a Barenblatt profile. Moreover, in the context of optimization of infinitely wide two layers neural networks [14, 26], $\mu_\star$ denotes the optimal distribution over the parameters of the network. In this setting, $F(x) = -2\int k(x,y)d\mu_\star(y)$ is non convex and $\mathcal{H}(\mu) = \int k(x,y)d\mu(x)d\mu(y)$ is an interaction energy [2, Example 9.3.4], with $k$ depending on the activation functions of the network. Moreover, $\mathcal{G}$ boils down to a Maximum Mean Discrepancy w.r.t. $\mu_\star$ [3] under a well-posedness condition. Alternatively, $F$ can be a regularizer of the distribution on the parameters of the network (see Appendix).

**Related works.** Wasserstein gradient flows are continuous time dynamics that can minimize (1). Several time discretizations of such flows have been considered [2, 33, 38]. However, these discretization schemes have been mainly analyzed as numerical schemes to approximate the continuous gradient flow, rather than as optimization algorithms.

Numerous optimization algorithms to solve (1), relying on different time-discretization schemes of the Wasserstein gradient flow of $\mathcal{G}$, have been proposed previously. For instance, [17, 33, 40] rely on the implementation of the JKO (Jordan-Kinderlehrer-Otto) scheme [19], which can be seen as a proximal scheme (*i.e.*, *backward discretization*) in the Wasserstein distance. In this case, each step of the algorithm relies on evaluating the JKO operator of $\mathcal{G}$, exploiting efficient subroutines for this operator. When $\mathcal{G}$ is smooth, some gradient descent algorithms over the $\mathcal{P}_2(\mathcal{X})$ (*i.e.*, *forward discretizations*) have also been proposed [13, 20, 21]. However, their analysis is notably challenging without further convexity assumptions on $\mathcal{G}$. For example [13], which tackles the resolution of the Wasserstein barycenter problem (a nonconvex problem) via gradient descent, considers Gaussian input distributions. Another time-discretization of Wasserstein gradient flows can be found in the Langevin Monte Carlo (LMC) algorithm in the sampling literature. LMC can be seen as a splitting algorithm involving a gradient step for $\mathcal{E}_F$ and an exact solving of the flow of $\mathcal{H}$ (*Forward-Flow discretization* [38]) to solve (1) in the case where $\mathcal{H}$ is the negative entropy. Indeed, the Wasserstein gradient flow of the negative entropy can be computed exactly in discrete time.[1] Several works provide non asymptotic analyses of LMC in discrete time, see [5, 12, 16, 22, 29, 35, 39] among others. However, the convergence rate of the LMC algorithm does not match the convergence rate of the associated continuous time dynamics and LMC is a biased algorithm.

A natural approach to minimize the sum of a smooth and a nonsmooth convex functions over a Hilbert space is to apply the proximal gradient algorithm [4], which implements a gradient step for the smooth term and a proximal step for the nonsmooth one. Indeed, many nonsmooth regularizers admit closed form proximity operators[2]. In this work we propose and analyze a splitting algorithm, the Forward-Backward (FB) scheme, to minimize the functional $\mathcal{G}$ over $\mathcal{P}_2(\mathcal{X})$. The FB scheme can be seen as an analogue to the proximal gradient algorithm in the Wasserstein space. This algorithm has already been studied (but no convergence rate has been established) in [38] in the particular case where $\mathcal{H}$ is the negative entropy. Notably, [38, Section F.1] explains why the FB scheme should be unbiased, unlike Forward-Flow discretizations like LMC.

More precisely, the proposed algorithm implements a forward (gradient step) for the smooth term $\mathcal{E}_F$ and relies on the JKO operator for the nonsmooth term $\mathcal{H}$ [19] only, which plays the role of a proximity operator. This approach provides an algorithm with lower iteration complexity compared to the "full" JKO scheme applied to $\mathcal{E}_F + \mathcal{H}$.

**Contributions.** In summary, the Wasserstein gradient flows to minimize (1) are well understood and modelize many machine learning algorithms. Discretized Wasserstein gradient flows have been

proposed but little is known about their minimization power. In this work, we propose a natural optimization algorithm to solve Problem (1), which is a Forward-Backward discretization of the Wasserstein gradient flow. This algorithm is a generalization of a discretization scheme proposed in [38]. Our main contribution is to prove non asymptotic rates for the proposed scheme, under the assumptions that $F$ is smooth and convex and that $\mathcal{H}$ is convex along the generalized geodesics defined by the Wasserstein distance. We show that the obtained rates fortunately match the ones of the proximal gradient algorithm over Hilbert spaces.

The remainder is organized as follows. In Section 2 we provide some background knowledge in optimal transport and gradient flows. In Section 3, we introduce the Forward-Backward Euler discretization scheme. We study the FB scheme as an optimization algorithm and present our main results, *i.e.*, non-asymptotic rates for the resolution of (1) in Section 4. In Section 5 we illustrate the performance of our algorithm for a simple sampling task. The convergence proofs are postponed to the appendix.

## 2 Preliminaries

In this section, we introduce the notations and recall fundamental definitions and properties on optimal transport and gradient flows that will be used throughout the paper.

### 2.1 Notations

In the sequel, $\mathcal{P}_2(\mathcal{X})$ is the space of probability measures $\mu$ on $\mathcal{X}$ with finite second order moment. Denote $\mathcal{B}(\mathcal{X})$ the Borelian $\sigma$-field over $\mathcal{X}$. For any $\mu \in \mathcal{P}_2(\mathcal{X})$, $L^2(\mu)$ is the space of functions $f : (\mathcal{X}, \mathcal{B}(\mathcal{X})) \to (\mathcal{X}, \mathcal{B}(\mathcal{X}))$ such that $\int \|f\|^2 d\mu < \infty$. Note that the identity map $I$ is an element of $L^2(\mu)$. For any $\mu \in \mathcal{P}_2(\mathcal{X})$, we denote by $\|\cdot\|_\mu$ and $\langle\cdot,\cdot\rangle_\mu$ respectively the norm and the inner product of the space $L^2(\mu)$. For any measures $\mu, \nu$, we write $\mu \ll \nu$ if $\mu$ is absolutely continuous with respect to $\nu$, and we denote $Leb$ the Lebesgue measure over $\mathcal{X}$. The set of regular distributions of the Wasserstein space is denoted by $\mathcal{P}_2^r(\mathcal{X}) := \{\mu \in \mathcal{P}_2(\mathcal{X}), \ \mu \ll Leb\}$. If $f, g : \mathcal{X} \to \mathcal{X}$, the composition $f \circ g$ of $g$ by $f$ is sometimes denoted $f(g)$.

### 2.2 Optimal transport

For every measurable map $T$ defined on $(\mathcal{X}, \mathcal{B}(\mathcal{X}))$ and for every $\mu \in \mathcal{P}_2(\mathcal{X})$, we denote $T_{\#}\mu$ the pushforward measure of $\mu$ by $T$ characterized by the 'transfer lemma', i.e.:

$$\int \phi(y) dT_{\#}\mu(y) = \int \phi(T(x)) d\mu(x) \quad \text{for any measurable and bounded function } \phi. \quad (2)$$

Consider the 2-Wasserstein distance defined for every $\mu, \nu \in \mathcal{P}_2(\mathcal{X})$ by

$$W^2(\mu, \nu) := \inf_{\upsilon \in \Gamma(\mu,\nu)} \int \|x - y\|^2 d\upsilon(x, y), \quad (3)$$

where $\Gamma(\mu, \nu)$ is the set of couplings between $\mu$ and $\nu$ [36], *i.e.* the set of nonnegative measures $\upsilon$ over $\mathcal{X} \times \mathcal{X}$ such that $P_{\#}\upsilon = \mu$ (resp. $Q_{\#}\upsilon = \nu$) where $P : (x, y) \mapsto x$ (resp. $Q : (x, y) \mapsto y$) is the projection onto the first (resp. second) component.

We now recall the well-known Brenier theorem [9], [2, Section 6.2.3].

**Theorem 1.** Let $\mu \in \mathcal{P}_2^r(\mathcal{X})$ and $\nu \in \mathcal{P}_2(\mathcal{X})$. Then,

1. There exists a unique minimizer $\upsilon$ of (3). Besides, there exists a uniquely determined $\mu$-almost everywhere (a.e.) map $T_\mu^\nu : \mathcal{X} \to \mathcal{X}$ such that $\upsilon = (I, T_\mu^\nu)_{\#}\mu$ where $(I, T_\mu^\nu) : (x, y) \mapsto (x, T_\mu^\nu(x))$. Finally, there exists a convex function $f : \mathcal{X} \to \mathbb{R}$ such that $T_\mu^\nu = \nabla f$ $\mu$-a.e.

2. As a corollary,

$$W^2(\mu, \nu) = \int \|x - \nabla f(x)\|^2 d\mu(x) = \inf_{T: \ T_{\#}\mu=\nu} \int \|x - T(x)\|^2 d\mu(x). \quad (4)$$

3. If $g : \mathcal{X} \to \mathbb{R}$ is convex, then $\nabla g$ is well defined $\mu$-a.e. and if $\nu = \nabla g_{\#}\mu$, then $T_\mu^\nu = \nabla g$ $\mu$-a.e.

4. If $\nu \in \mathcal{P}_2^r(\mathcal{X})$, then $T_\nu^\mu \circ T_\mu^\nu = I$ $\mu$-a.e. and $T_\mu^\nu \circ T_\nu^\mu = I$ $\nu$-a.e.

Under the assumptions of Theorem 1, the map $T_\mu^\nu$ is called the optimal transport (OT) map from $\mu$ to $\nu$. In this paper, as it is commonly the case in the literature, we may refer to the space of probability distributions $\mathcal{P}_2(\mathcal{X})$ equipped with the 2-Wasserstein distance as the Wasserstein space.

## 2.3 Review of Gradient Flows and their discretizations

### 2.3.1 In an Euclidean space

Assume that $\mathcal{X}$ is a Euclidean space, consider a proper lower semi-continuous function $G : \mathcal{X} \to (-\infty, +\infty]$ and denote by $D(G) = \{x \in \mathcal{X}, \ G(x) < \infty\}$ its domain. We assume that $G$ is convex, *i.e.*, for every $x, z \in \mathcal{X}$ and for every $\varepsilon \in [0, 1]$, we have:

$$G(\varepsilon z + (1 - \varepsilon)x) \le \varepsilon G(z) + (1 - \varepsilon)G(x). \qquad (5)$$

Given $x \in \mathcal{X}$, recall that $y \in \mathcal{X}$ is a subgradient of $G$ at $x$ if for every $z \in \mathcal{X}$,

$$G(x) + \langle y, z - x \rangle \le G(z).$$

The (possibly empty) set of subgradients of $G$ at $x$ is denoted by $\partial G(x)$, and the map $x \mapsto \partial G(x)$ is called the subdifferential. If $G$ is differentiable at $x$, then $\partial G(x) = \{\nabla G(x)\}$ where $\nabla G(x)$ is the gradient of $G$ at $x$. The subdifferential of the convex function $G$ allows to define the gradient flow of $G$: for every initial condition $\mathsf{x}(0) = a$ such that $\partial G(a) \ne \emptyset$, there exists a unique absolutely continuous function $\mathsf{x} : [0, +\infty) \to \mathcal{X}$ that solves the differential inclusion [10, Th. 3.1], [30, Th. 2.7]

$$\mathsf{x}'(t) \in \partial G(\mathsf{x}(t)). \qquad (6)$$

One can check that the gradient flow of $G$ is also characterized by the following system of Evolution Variational Inequalities (EVI) :

$$\forall z \in D(G), \quad \frac{d}{dt}\|\mathsf{x}(t) - z\|^2 \le -2\left(G(\mathsf{x}(t)) - G(z)\right).$$

In contrast to (6), the former characterization allows to define the gradient flow without using the notion of subdifferential, a property that can be practical in nonsmooth settings. Moreover, the non-asymptotic analysis of discretized gradient flows in the optimization literature often relies on discrete versions of the EVI.

The existence of Gradient Flows can be established as the limit of a proximal scheme [30, Th. 2.14], [6, Th. 5.1] when the step-size $\gamma \to 0$. Defining the proximity operator of $G$ as:

$$\operatorname{prox}_{\gamma G}(x) := \underset{y \in \mathcal{X}}{\arg\min}\, G(y) + \frac{1}{2\gamma}\|y - x\|^2, \qquad (7)$$

the proximal scheme is written as

$$x_{n+1} = \operatorname{prox}_{\gamma G}(x_n), \qquad (8)$$

which corresponds to the proximal point algorithm to minimize the function $G$, see [23]. The proximal scheme can be seen as a *Backward Euler discretization* of the gradient flow. Indeed, writing the first order conditions of (8), we have

$$x_{n+1} \in x_n - \gamma \partial G(x_{n+1}), \quad \text{or equivalently} \quad \frac{x_{n+1} - x_n}{\gamma} \in -\partial G(x_{n+1}).$$

Hence, each iteration of the proximal scheme requires solving an equation which can be intractable in many cases. The *Forward Euler scheme* is a more tractable integrator of the gradient flow of $G$, but is less stable and requires the differentiability of $G$. Under this assumption, this scheme is written

$$\frac{x_{n+1} - x_n}{\gamma} = -\nabla G(x_n) \quad \text{or equivalently} \quad x_{n+1} = x_n - \gamma \nabla G(x_n), \qquad (9)$$

which corresponds to the well-known gradient descent algorithm to minimize the function $G$. Consider now the case where the function $G$ can be decomposed as $G = F + H$, where $F$ is convex and smooth

and $H$ is convex and nonsmooth. To integrate the gradient flow of $G = F + H$, another approach is to use the Forward and the Backward Euler schemes for the smooth term and nonsmooth term respectively [30]. This approach is also motivated by the fact that in many situations, the function $H$ is simple enough to implement its proximity operator $\text{prox}_{\gamma H}$. If $G = F + H$, the Forward Backward Euler scheme is written as

$$\frac{x_{n+1} - x_n}{\gamma} \in -\nabla F(x_n) - \partial H(x_{n+1}). \tag{10}$$

Recalling the definition of the proximity operator, this scheme can be rewritten as

$$x_{n+1} = \text{prox}_{\gamma H}(x_n - \gamma \nabla F(x_n)), \tag{11}$$

which corresponds to the proximal gradient algorithm to minimize the composite function $G$.

### 2.3.2 In the Wasserstein space

Consider a proper lower semi-continuous functional $\mathcal{G} : \mathcal{P}_2(\mathcal{X}) \to (-\infty, +\infty]$ and denote $D(\mathcal{G}) = \{\mu \in \mathcal{P}_2(\mathcal{X}), \ \mathcal{G}(\mu) < \infty\}$ its domain. We assume that $\mathcal{G}$ is convex along generalized geodesics defined by the 2-Wasserstein distance [2, Chap. 9], *i.e.* for every $\mu, \pi \in \mathcal{P}_2(\mathcal{X}), \nu \in \mathcal{P}_2^r(\mathcal{X})$ and for every $\varepsilon \in [0, 1]$,

$$\mathcal{G}((\varepsilon T_\nu^\pi + (1 - \varepsilon) T_\nu^\mu)_{\#}\nu) \le \varepsilon \mathcal{G}(\pi) + (1 - \varepsilon)\mathcal{G}(\mu), \tag{12}$$

where $T_\nu^\pi$ and $T_\nu^\mu$ are the optimal transport maps from $\nu$ to $\pi$ and from $\nu$ to $\mu$ respectively. Note that the curve $\varepsilon \mapsto (\varepsilon T_\nu^\pi + (1 - \varepsilon) T_\nu^\mu)_{\#}\nu$ interpolates between $\mu$ ($\varepsilon = 0$) and $\pi$ ($\varepsilon = 1$). Moreover, if $\nu = \pi$ or $\nu = \mu$, then this curve is a geodesic in the Wasserstein space. Given $\mu \in \mathcal{P}_2(\mathcal{X}), \xi \in L^2(\mu)$ is a strong Fréchet subgradient of $\mathcal{G}$ at $\mu$ [2, Chap. 10] if for every $\phi \in L^2(\mu)$,

$$\mathcal{G}(\mu) + \varepsilon \langle \xi, \phi \rangle_\mu + o(\varepsilon) \le \mathcal{G}((I + \varepsilon \phi)_{\#}\mu).$$

The (possibly empty) set of strong Fréchet subgradients of $\mathcal{G}$ at $\mu$ is denoted $\partial \mathcal{G}(\mu)$. The map $\mu \mapsto \partial \mathcal{G}(\mu)$ is called the strong Fréchet subdifferential. Conveniently, the strong Fréchet subdifferential enables to define the gradient flow of the functional $\mathcal{G}$ [2, Chap. 11]. However in the nonsmooth setting that will be considered in this paper, the characterization of gradient flows through EVI will be more practical. The gradient flow of $\mathcal{G}$ is the solution of the following system of EVI [2, Th. 11.1.4]:

$$\forall \pi \in D(\mathcal{G}), \quad \frac{d}{dt} W^2(\mu(t), \pi) \le -2 \left( \mathcal{G}(\mu(t)) - \mathcal{G}(\pi) \right).$$

We shall perform a non-asymptotic analysis of a discretized gradient flow scheme to minimize the functional $\mathcal{G}$. Our approach is to prove a discrete EVI for this scheme.

The existence of gradient flows can be established as the limit of a minimizing movement scheme [2, Th. 11.2.1], [19]. Defining the JKO operator of $\mathcal{G}$ as:

$$\text{JKO}_{\gamma \mathcal{G}}(\mu) := \underset{\nu \in \mathcal{P}_2(\mathcal{X})}{\arg\min} \ \mathcal{G}(\nu) + \frac{1}{2\gamma} W^2(\nu, \mu), \tag{13}$$

the JKO scheme is written as

$$\mu_{n+1} \in \text{JKO}_{\gamma \mathcal{G}}(\mu_n).$$

The JKO operator can be seen as a proximity operator by replacing the Wasserstein distance by the Euclidean distance. Moreover, the JKO scheme can be seen as a Backward Euler discretization of the gradient flow. Indeed, under some assumptions on the functional $\mathcal{G}$, using [2, Lemma 10.1.2] we have

$$\frac{T_{\mu_{n+1}}^{\mu_n} - I}{\gamma} \in \partial \mathcal{G}(\mu_{n+1}).$$

Using Brenier's theorem, since $T_{\mu_{n+1}}^{\mu_n} \circ T_{\mu_n}^{\mu_{n+1}} = I \ \mu_n$-a.e., there exists a strong Fréchet subgradient of $\mathcal{G}$ at $\mu_{n+1}$ denoted by $\nabla_W \mathcal{G}(\mu_{n+1})$ such that

$$\mu_{n+1} = \left( I - \gamma \nabla_W \mathcal{G}(\mu_{n+1}) \circ T_{\mu_n}^{\mu_{n+1}} \right)_{\#} \mu_n.$$

Each iteration of the JKO scheme thus requires the minimization of a function which can be intractable in many cases. As previously, the Forward Euler scheme is more tractable and enjoys additionally

a simpler geometrical interpretation. Assume $\partial\mathcal{G}(\mu) = \{\nabla\mathcal{G}(\mu)\}$ is a singleton for any $\mu \in D(\mathcal{G})$ (some examples are given [2, Section 10.4]). The Forward Euler scheme for the gradient flow of $\mathcal{G}$ is written:

$$\mu_{n+1} = (I - \gamma\nabla\mathcal{G}(\mu_n))_{\#}\mu_n, \tag{14}$$

and corresponds to the iterations of the gradient descent algorithm over the Wasserstein space to minimize $\mathcal{G}$. Although the Wasserstein space is not a Riemannian manifold, it can still be equipped with a Riemannian structure and interpretation [25, 28]. In particular, the Forward Euler scheme can be seen as a Riemannian gradient descent where the exponential map at $\mu$ is the map $\phi \mapsto (I + \phi)_{\#}\mu$ defined on $L^2(\mu)$.

# 3   The Forward Backward Euler scheme

Recall that our goal is to minimize $\mathcal{E}_F + \mathcal{H}$, where $\mathcal{E}_F(\mu) = \int F(x)d\mu(x)$ for any $\mu \in \mathcal{P}_2(\mathcal{X})$, and $\mathcal{H}$ is a nonsmooth functional. Throughout this paper, we assume the following on the potential function $F$: there exists $L, \lambda \geq 0$ such that

- **A1.** $F$ is $L$-smooth *i.e.* $F$ is differentiable and $\nabla F$ is $L$-Lipschitz continuous; for all $(x, y) \in \mathcal{X}^2$:

$$F(y) \leq F(x) + \langle\nabla F(x), y - x\rangle + \frac{L}{2}\|x - y\|^2. \tag{15}$$

- **A2.** $F$ is $\lambda$-strongly convex (we allow $\lambda = 0$); for all $(x, y) \in \mathcal{X}^2$:

$$F(x) \leq F(y) - \langle\nabla F(x), y - x\rangle - \frac{\lambda}{2}\|x - y\|^2. \tag{16}$$

Moreover, we assume the following on the function $\mathcal{H}$.

- **B1.** $\mathcal{H} : \mathcal{P}_2(\mathcal{X}) \to (-\infty, +\infty]$ is proper and lower semi-continuous. Moreover, $D(\mathcal{H}) \subset \mathcal{P}_2^r(\mathcal{X})$.
- **B2.** There exists $\gamma_0 > 0$ such that $\forall\gamma \in (0, \gamma_0)$, $\mathrm{JKO}_{\gamma\mathcal{H}}(\mu) \neq \emptyset$ for every $\mu \in \mathcal{P}_2(\mathcal{X})$.
- **B3.** $\mathcal{H}$ is convex along generalized geodesics.

Assumptions **B1** and **B2** are general technical assumptions, used in [2] (see [2, Eq. 10.1.1a, 10.1.1b]) that are satisfied in relevant cases. Then [2, Prop. 9.3.2, 9.3.5 and 9.3.9] gives broad examples of (potential, interaction and internal) energies satisfying **B3**, *e.g.*, potential (resp. interaction) energies if the potential (resp. interaction) term is convex, and entropies (see Appendix and [2, Remark 9.3.10]). Moreover, $\mathcal{H}$ is often nonsmooth, see *e.g.* [38, Section 3.1.1]. Therefore, we use a Forward Backward Euler scheme to integrate the gradient flow of $\mathcal{G}$. Let $\gamma > 0$ a step size. The proposed Forward Backward Euler scheme is written, for $n \geq 0$, as:

$$\nu_{n+1} = (I - \gamma\nabla F)_{\#}\mu_n \tag{17}$$
$$\mu_{n+1} \in \mathrm{JKO}_{\gamma\mathcal{H}}(\nu_{n+1}). \tag{18}$$

This scheme can be seen as a proximal gradient algorithm over the Wasserstein space to minimize the composite function $\mathcal{G} = \mathcal{E}_F + \mathcal{H}$.

**Remark 1.** To our knowledge, the two cases where $\mathrm{JKO}_{\gamma\mathcal{H}}$ can be computed in closed form are the case where $\mathcal{H}(\mu) = \int R d\mu$ and $R$ is a proximable function [3], see [8]; and the case where $\mathcal{H}$ is the negative entropy and $F$ is quadratic, see [38, Example 8] and Section 5. However, there exist subroutines to compute JKOs of generic functionals, as well documented in the optimal transport and PDE literature (see the review of different strategies in [33]). The implementation of the FB scheme relies on these subroutines, similarly to proximal splitting algorithms in optimization that rely on subroutines to compute the proximity operator. Moreover, as several proximity operators admit a close form[4] one can hope to be able to compute $\mathrm{JKO}_{\gamma\mathcal{H}}$ for simple functionals $\mathcal{H}$.

For instance, when $\mathcal{H}$ is the negative entropy (defined by $\mathcal{H}(\mu) = \int \log\left(\mu(x)\right)d\mu(x)$ if $\mu \ll Leb$ with density $\mu$ and $\mathcal{H}(\mu) := +\infty$ else), we conjecture that a technique similar to [18] can be

applied. Again, for this particular functional, the JKO is known in closed form in the Gaussian case [38, Example 8]. Other works of interest for the proposed FB scheme have investigated efficient methods to implement the JKO of a generic functional with respect to the entropy-regularized Wasserstein distance [31].

In the next section, we study the non asymptotic properties of the FB scheme.

# 4 Non asymptotic analysis

In this section, we provide rates for the Wasserstein proximal gradient algorithm. The main technical challenge is to handle the fact that $\mathcal{H}$ is only convex along generalized geodesics (and not along any interpolating curve). We overcome this challenge by using the intuition that the Wasserstein proximal gradient algorithm is a discretization of the associated Wasserstein gradient flow. More precisely, the main step is to establish a discrete EVI to prove the rates. The proof differs from the one of the proximal gradient algorithm because the convexity inequality can be used with optimal transport maps only, see Lemma 4.

We consider a fixed step size $\gamma < 1/L$ and a probability distribution $\pi \in \mathcal{P}_2(\mathcal{X})$. Our main result (Proposition 8) combines several ingredients: the identification of the optimal transport maps between $\mu_n, \nu_{n+1}$ and $\mu_{n+1}$ (see Equations (17) and (18)), the proof of a generic lemma regarding generalized geodesic convexity (Lemma 4) and a proof of a discrete EVI (Lemma 6).

## 4.1 Identification of optimal transport maps

Lemmas 2,3 identify the optimal transport maps from $\mu_n$ to $\nu_{n+1}$ and from $\nu_{n+1}$ to $\mu_{n+1}$ in the Forward Backward Euler scheme, as soon as the step size is sufficiently small. In particular, Lemma 3 is a consequence of [2, Lemma 10.1.2].

**Lemma 2.** Assume **A1**. Let $\mu \in \mathcal{P}_2^r(\mathcal{X})$ and $\nu = (I - \gamma \nabla F)_{\#}\mu$. Then if $\gamma < 1/L$, the optimal transport map from $\mu$ to $\nu$ corresponds to

$$T_\mu^\nu = I - \gamma \nabla F.$$

Moreover, $\nu$ belongs to $\mathcal{P}_2^r(\mathcal{X})$.

**Lemma 3.** Assume **B1, B2**. Let $\nu \in \mathcal{P}_2(\mathcal{X})$. If $\mu \in \mathrm{JKO}_{\gamma\mathcal{H}}(\nu)$, then $\mu \in D(\mathcal{H}) \subset \mathcal{P}_2^r(\mathcal{X})$ and the optimal transport map $T_\mu^\nu$ from $\mu$ to $\nu$ satisfies $T_\mu^\nu \in I + \gamma \partial \mathcal{H}(\mu)$. In other words, there exists a strong Fréchet subgradient at $\mu$ denoted by $\nabla_W \mathcal{H}(\mu)$ such that

$$T_\mu^\nu = I + \gamma \nabla_W \mathcal{H}(\mu). \tag{19}$$

Using Lemmas 2,3, if $\mu_0 \in \mathcal{P}_2^r(\mathcal{X})$, then $\mu_n, \nu_n \in \mathcal{P}_2^r(\mathcal{X})$ for every $n$ by induction. This remark allows to consider optimal transport maps from $\mu_n$ and $\nu_n$ to any $\pi \in \mathcal{P}_2(\mathcal{X})$. The next lemma extends [2, 10.1.1.B] to functionals $\mathcal{H}$ convex along generalized geodesics.

**Lemma 4.** Assume **B1, B2, B3**. Let $\nu \in \mathcal{P}_2^r(\mathcal{X})$, $\mu, \pi \in \mathcal{P}_2(\mathcal{X})$ and $T_\nu^\mu, T_\nu^\pi$ the optimal transport maps from $\nu$ to $\mu$ and from $\nu$ to $\pi$ respectively. If $\xi \in \partial \mathcal{H}(\mu)$, then

$$\langle \xi \circ T_\nu^\mu, T_\nu^\pi - T_\nu^\mu \rangle_\nu \leq \mathcal{H}(\pi) - \mathcal{H}(\mu). \tag{20}$$

Lemma 4 is natural, holds for any functional convex along generalized geodesics, and it is novel, to the best of our knowledge. The following results rely on this lemma.

## 4.2 A descent lemma

Without using any convexity assumption on $F$, we first obtain a descent lemma. We denote $Y_{n+1} := T_{\nu_{n+1}}^{\mu_{n+1}}$ the optimal transport map between $\nu_{n+1}$ and $\mu_{n+1}$ in the Forward Backward Euler scheme (17), (18), and $X_{n+1} := Y_{n+1} \circ (I - \gamma \nabla F)$. Note that $X_{n+1}$ is a pushforward from $\mu_n$ to $\mu_{n+1}$.

**Theorem 5** (Descent). Assume $\mu_0 \in \mathcal{P}_2^r(\mathcal{X})$, $\gamma < 1/L$ and **A1, B1, B2, B3**. Then for $n \geq 0$, there exists a strong Fréchet subgradient at $\mu_{n+1}$ denoted by $\nabla_W \mathcal{H}(\mu_{n+1})$ such that:

$$\mathcal{G}(\mu_{n+1}) \leq \mathcal{G}(\mu_n) - \gamma \left(1 - \frac{L\gamma}{2}\right) \|\nabla F + \nabla_W \mathcal{H}(\mu_{n+1})(X_{n+1})\|_{\mu_n}^2,$$

where we use the notation $\nabla_W \mathcal{H}(\mu_{n+1})(X_{n+1})$ to denote $\nabla_W \mathcal{H}(\mu_{n+1}) \circ X_{n+1}$.

Hence, the sequence $(\mathcal{G}(\mu_n))_n$ is decreasing as soon as the step-size is small enough.

### 4.3 Discrete EVI

To prove a discrete EVI and obtain convergence rates, we need the additional convexity assumption **A2** on the potential function $F$. We firstly prove the two following lemmas.

**Lemma 6.** Assume $\mu_0 \in \mathcal{P}_2^r(\mathcal{X})$, $\gamma < 1/L$, and **B1, B2, B3**. Then for $n \geq 0$ and $\pi \in \mathcal{P}_2(\mathcal{X})$, there exists a strong Fréchet subgradient at $\mu_{n+1}$ denoted $\nabla_W \mathcal{H}(\mu_{n+1})$ such that:

$$W^2(\mu_{n+1}, \pi) \leq W^2(\nu_{n+1}, \pi) - 2\gamma \left( \mathcal{H}(\mu_{n+1}) - \mathcal{H}(\pi) \right) - \gamma^2 \|\nabla_W \mathcal{H}(\mu_{n+1})\|_{\mu_{n+1}}^2 .$$

**Lemma 7.** Assume $\mu_0 \in \mathcal{P}_2^r(\mathcal{X})$, $\gamma \leq 1/L$, and **A1, A2** with $\lambda \geq 0$. Then for $n \geq 0$, and $\pi \in \mathcal{P}_2(\mathcal{X})$

$$W^2(\nu_{n+1}, \pi) \leq (1 - \gamma\lambda) W^2(\mu_n, \pi) - 2\gamma \left( \mathcal{E}_F(\mu_n) - \mathcal{E}_F(\pi) \right) + \gamma^2 \|\nabla F\|_{\mu_n}^2 .$$

We can now provide a discrete EVI for the functional $\mathcal{G} = \mathcal{E}_F + \mathcal{H}$.

**Proposition 8** (discrete EVI). Assume $\mu_0 \in \mathcal{P}_2^r(\mathcal{X})$, $\gamma < 1/L$, and **A1–B3** with $\lambda \geq 0$. Then for $n \geq 0$ and $\pi \in \mathcal{P}_2(\mathcal{X})$, there exists a strong Fréchet subgradient at $\mu_{n+1}$ denoted by $\nabla_W \mathcal{H}(\mu_{n+1})$ such that the Forward Backward Euler scheme verifies:

$$W^2(\mu_{n+1}, \pi) \leq (1 - \gamma\lambda) W^2(\mu_n, \pi) - 2\gamma \left( \mathcal{G}(\mu_{n+1}) - \mathcal{G}(\pi) \right). \tag{21}$$

### 4.4 Convergence rates

When the potential function $F$ is convex, we easily get rates from the discrete EVI inequality provided above. Theorem 9 is a direct consequence of Proposition 8 by taking $\pi = \mu_\star$, and its corollaries provide rates depending on the strong convexity parameter of $F$.

**Theorem 9.** Assume $\mu_0 \in \mathcal{P}_2^r(\mathcal{X})$, $\gamma < 1/L$, and **A1–B3** with $\lambda \geq 0$. Then for every $n \geq 0$,

$$W^2(\mu_{n+1}, \mu_\star) \leq (1 - \gamma\lambda) W^2(\mu_n, \mu_\star) - 2\gamma (\mathcal{G}(\mu_{n+1}) - \mathcal{G}(\mu_\star)).$$

**Corollary 10** (Convex case rate). Under the assumptions of Theorem 9, for $n \geq 0$:

$$\mathcal{G}(\mu_n) - \mathcal{G}(\mu_\star) \leq \frac{W^2(\mu_0, \mu_\star)}{2\gamma n}.$$

**Corollary 11** (Strongly convex case rate). Under the assumptions of Theorem 9, if $\lambda > 0$, then for $n \geq 0$:

$$W^2(\mu_n, \mu_\star) \leq (1 - \gamma\lambda)^n W^2(\mu_0, \mu_\star).$$

Hence, as soon as $F$ is convex, we get sublinear rates in terms of the objective function $\mathcal{G}$, while when $F$ is $\lambda$-strongly convex with $\lambda > 0$, we get linear rates in the squared Wasserstein distance for the iterates of the Forward Backward Euler scheme. These rates match those of the proximal gradient algorithm in Hilbert space in the convex and strongly convex cases. [27]. Moreover, these rates are discrete time analogues of the continuous time rates obtained in [2, Th. 11.2.1] for the gradient flow of $\mathcal{G}$. In the particular case where $\mathcal{H}$ is the negative entropy, we can compare the convergence rates of the FB scheme to those of LMC. Although, the complexity of one iteration of the FB scheme is higher, the convergence rates of the FB scheme are faster, see *e.g.* [16].

## 5 Numerical experiments

We provide numerical experiments with a ground truth target distribution $\mu_\star$ to illustrate the dynamical behavior of the FB scheme, similarly to [34, Section 4.1]. We consider $F(x) = 0.5|x|^2$, and $\mathcal{H}$ the negative entropy. In this case, $\mathcal{G}(\mu)$ is (up to an additive constant) the Kullback-Leibler divergence of $\mu$ w.r.t. the standard Gaussian distribution $\mu_\star$. We denote by $m_\star$ the mean and $\sigma_\star$ the variance, and fix $m_\star = 0$ and $\sigma_\star^2 = 1.0$. We use the closed-form particle implementation of the FB scheme [38, Section G.1]. This allows to show the dynamical behavior of the FB scheme when $\gamma = 0.1$, and $\mu_0$ is Gaussian with $m_0 = 10$ and $\sigma_0 = 100$, in Figure 1. Note that $\lambda = 1.0$.

More precisely, the position of a set of particles initially distributed according to $\mu_0$ is updated iteratively. Using [38, Example 8], if $\mu_\star$ and $\mu_0$ are Gaussian, then $\mu_n$ is Gaussian for every $n$. Moreover, the mean (resp. the covariance matrix) of $\mu_{n+1}$ can be expressed as a function of the

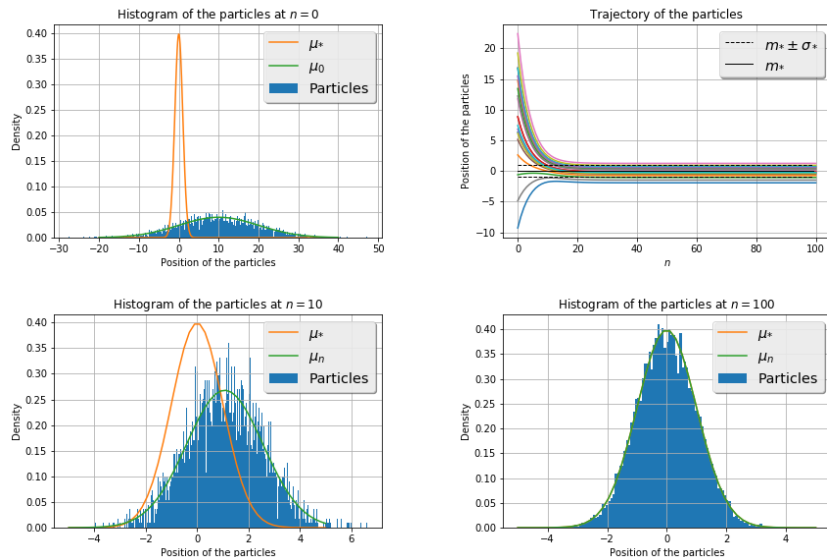

Figure 1: The particle implementation of the FB scheme illustrate the convergence of $\mu_n$ to $\mu_\star$.

mean (resp. the covariance matrix) of $\mu_n$. Finally, the update of the position of the particles can be computed using these expressions, see the update formulas in [38, Example 8]. These expressions also allow to compute $W^2(\mu_n, \mu_\star)$: since $\mu_n$ and $\mu_\star$ are Gaussian, $W^2(\mu_n, \mu_\star)$ can be computed in closed form and is also known as Bures-Wasserstein distance.

The empirical distribution of the particles, represented by histograms, approximates $\mu_n$. We see that $\mu_n$ matches $\mu_\star$ after few iterations on Figure 1. We also illustrate the particles, from their initial position (distributed according to $\mu_0$) to their final position (distributed according to $\mu_{100}$) in a high probability region of the target distribution $\mu_\star$. This shows that $\mu_{100}$ is close to $\mu_\star$. The linear convergence of $\mu_n$ to $\mu_\star$ in the squared Wasserstein distance is illustrated in a multidimensional case in Figure 2 (see Appendix).

# 6    Conclusion

We proposed an unified analysis of a Forward-Backward discretization of Wasserstein gradient flows in the convex case. We showed that the Forward-Backward discretization has convergence rates that are similar to the ones of the proximal gradient algorithm in Euclidean spaces.

Note that the implementation of the JKO operators is independent from the analysis of the FB scheme. However, an important problem raised by our work is whether we can find efficient implementations of the JKO operators of some specific functionals $\mathcal{H}$ relevant in machine learning applications.

# 7    Acknowledgement

Adil Salim thanks Pascal Bianchi for suggesting to use the JKO operator for optimization purposes.

# 8    Broader impact

The results that we showed, together with efficient implementations of some *specific* JKOs could be very impactful for many machine learning tasks.

## Footnotes

[1]The addition of a gaussian noise corresponds to a process whose distribution is solution of the gradient flow of the relative entropy (the heat flow).

[2]see www.proximity-operator.net

[3]*i.e.*, $R$ is a lower semi-continuous proper convex function whose proximity operator has a closed form

[4]see www.proximity-operator.net

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
