[Supplementary Material]

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

# A    Further numerical experiment

We consider solving Problem (1), in a case where the ground truth $\mu_\star$ is a high dimensional Gaussian distribution [13].

We shall illustrate the linear convergence of the FB scheme this problem.

We consider a multidimensional extension of the simulation of Section 5. More precisely, $F(x) = 0.5\|x\|^2$ and, using the notations of Section 5, the inital distribution is $\mu_0^{\otimes d}$ and the target distribution is $\mu_\star^{\otimes d}$, where $\otimes$ denotes the product of measures.

The mean and covariance matrix of $\mu_n$ can be computed in closed form using [38, Example 8], therefore we can also compute $W^2(\mu_n, \mu_\star)$.

The simulation can no longer be represented with histograms and particles, however, we represent the linear convergence of $\mu_n$ to $\mu_\star$ predicted by Corollary 11 in Figure 2.

Figure 2: Linear convergence of $W^2(\mu_n, \mu_\star)$ to 0 in dimension $d = 1000$.

We observe that $\mu_n$ converges linearly to $\mu_\star$ as predicted. Moreover, the observed linear rate is better than the one predicted by our theory. This suggests that our results can be improved in some particular cases.

# B    Examples of functionals $\mathcal{H}$

## B.1    Internal energies

An internal energy is written

$$\mathcal{H}(\mu) = \int h(\mu(x))dx, \tag{22}$$

if $\mu \ll Leb$ with density $\mu(x)$, and $\mathcal{H}(\mu) = +\infty$ else. Internal energies satisfying Assumptions **B1**, **B2**, **B3** include the negative entropy, which corresponds to $h(u) = u \log(u)$, and the higher order entropies, which correspond to $h(u) = u^v/v$ where $v > 1$, see [2, Remark 9.3.10]. Moreover,

in the case of the negative entropy, there is only one minimizer $\mu_\star$ of Problem (1), and its density w.r.t. $Leb$ can be written $\exp(C - F(x))$, where $C \in \mathbb{R}$. In the case of the higher order entropies, $\mu_\star(x) = \max(0, C - F(x))^{\frac{1}{v-1}}$.

## B.2 [Nonconvex] Neural network optimization

Consider a infinite-width one hidden layer neural network (1HLNN) [3, 14, 26] and let $n$ be the number of neurons. There are several ways to cast the optimization of a 1HLNN as Problem (1).

For any input $x$, the output of the 1HLNN can be written as $f(x) = \frac{1}{n} \sum_{i=1}^{n} \phi(x, z_i)$, where $\phi(x, z_i) = w_i \psi(x, \theta_i)$ with $w_i \in \mathbb{R}$ the weight of the $i$-th neuron and $\theta_i \in \mathbb{R}^d$ parametrizes the activation function $\psi$ (e.g. a sigmoid). Given data $(x, y) \sim p$, the optimization of this neural network can be written as the minimization of the risk: $\arg\min_{z_1, \dots, z_n \in \mathsf{Z}} \mathbb{E}_{(x,y) \sim p}[\ell\left(y, \frac{1}{n} \sum_{i=1}^{n} \phi(x, z_i)\right)]$, where $\mathsf{Z} = \mathbb{R} \times \mathbb{R}^d$ and $\ell$ is some loss function. When $n \to \infty$, this becomes an optimization problem over $\mathcal{P}_2(\mathsf{Z})$: $\arg\min_{\mu \in \mathcal{P}(\mathsf{Z})} \mathbb{E}_{(x,y) \sim p}[\ell\left(y, \int \phi(x, z) d\mu(z)\right)]$. Denote $R(h) = \mathbb{E}_{(x,y) \sim p}[\ell\left(y, h(x)\right)]$.

In [14], the optimization of a 1HLNN is cast as Problem (1), where $F$ is a regularizer on the parameters of the network (e.g. $F(x) = \|x\|^2$) and $\mathcal{H}(\mu) = R\left(\int \phi d\mu\right)$ is the risk.

Alternatively, consider the case where $\ell(y, x) = \frac{1}{2}\|y - x\|^2$ as in [3]. Expanding the risk function leads to the objective $\mathcal{G}(\mu) = \int F(z) d\mu(z) + \frac{1}{2} \int K(z, z') d\mu(z) d\mu(z')$, where $F(z) = -\mathbb{E}_{(x,y) \sim p}[y\phi(x, z)]$ is a potential and $K(z, z') = \mathbb{E}_{x \sim p_x}[\phi(x, z)\phi(x, z')]$ is an interaction term.

Note however that optimizing a 1HLNN is in general only a geodesically semiconvex problem [3, 14] (which is weaker than (12), see [2, Definition 9.1.1]), and hence is not strictly covered by our theory.

## C    Proof of Lemma 2

The map $I - \gamma\nabla F$ is a pushforward from $\mu$ to $\nu$. Moreover, denoting $u : x \mapsto \frac{1}{2}\|x\|^2 - \gamma F(x)$, $\nabla u = I - \gamma \nabla F$.

By elementary algebra, for any $(x, y) \in \mathcal{X}^2$ we have:

$$\frac{1}{2}\|x\|^2 = \frac{1}{2}\|y\|^2 - \langle x, y - x \rangle - \frac{1}{2}\|x - y\|^2, \tag{23}$$

and from the smoothness of $F$,

$$F(y) \le F(x) + \langle \nabla F(x), y - x \rangle + \frac{L}{2}\|x - y\|^2. \tag{24}$$

Therefore, combining (23) and (24) multiplied by $\gamma$ gives:

$$\frac{1}{2}\|x\|^2 - \gamma F(x) \le \frac{1}{2}\|y\|^2 - \gamma F(y) - \langle x - \gamma\nabla F(x), y - x \rangle - \frac{1}{2}(1 - L\gamma)\|x - y\|^2. \tag{25}$$

In other words,

$$u(x) \le u(y) - \langle \nabla u(x), y - x \rangle - \frac{1}{2}(1 - L\gamma)\|x - y\|^2. \tag{26}$$

Therefore, if $\gamma \le 1/L$, then $u$ is convex and $\nabla u = T_\mu^\nu$ using Brenier's theorem. Moreover, if $\gamma < 1/L$ then $u$ is $(1 - L\gamma)$-strongly convex. In consequence,

$$(1 - L\gamma)\|x - y\|^2 \le \langle x - y, \nabla u(x) - \nabla u(y) \rangle.$$

Therefore, $\nabla u$ is injective. Furthermore, using the strong convexity of $u$ and [2, Lemma 5.5.3] (see also [2, Th. 6.2.3, Th. 6.2.7]), $\nu \in \mathcal{P}_2^r(\mathcal{X})$.

## D    Proof of Lemma 3

Let $\mu \in \mathrm{JKO}_{\gamma\mathcal{H}}(\nu)$. Since $D(\mathcal{H}) \subset \mathcal{P}_2^r(\mathcal{X})$, [2, Lemma 10.1.2] implies $\mu \in D(\mathcal{H})$ and $\frac{1}{\gamma}(T_\mu^\nu - I) \in \partial\mathcal{H}(\nu)$ is a strong subgradient of $\mathcal{H}$ at $\nu$.

# E  Proof of Lemma 4

Since $\xi \in \partial\mathcal{H}(\mu)$, for every $\phi \in L^2(\mu)$,

$$\mathcal{H}((I + \varepsilon\phi)_{\#}\mu) \geq \mathcal{H}(\mu) + \varepsilon\langle\xi, \phi\rangle_\mu + o(\varepsilon).$$

Applying the last inequality to $\phi = T_\nu^\pi \circ T_\mu^\nu - I$ and using the transfer lemma ($\mu = T_{\nu}^{\mu}{}_{\#}\nu$) we have

$$\mathcal{H}((T_\nu^\mu + \varepsilon(T_\nu^\pi - T_\nu^\mu))_{\#}\nu) = \mathcal{H}((I + \varepsilon\phi)_{\#}\mu),$$

$$\langle\xi, \phi\rangle_\mu = \langle\xi(T_\nu^\mu), T_\nu^\pi - T_\nu^\mu\rangle_\nu,$$

and

$$\frac{\mathcal{H}((T_\nu^\mu + \varepsilon(T_\nu^\pi - T_\nu^\mu))_{\#}\nu) - \mathcal{H}(\mu)}{\varepsilon} \geq \langle\xi(T_\nu^\mu), T_\nu^\pi - T_\nu^\mu\rangle_\nu + o(1). \tag{27}$$

Using the generalized geodesic convexity of $\mathcal{H}$,

$$\mathcal{H}((T_\nu^\mu + \varepsilon(T_\nu^\pi - T_\nu^\mu))_{\#}\nu) \leq \varepsilon\mathcal{H}(\pi) + (1 - \varepsilon)\mathcal{H}(\mu).$$

Plugging the last inequality into (27),

$$\mathcal{H}(\pi) - \mathcal{H}(\mu) \geq \langle\xi(T_\nu^\mu), T_\nu^\pi - T_\nu^\mu\rangle_\nu + o(1). \tag{28}$$

We get the conclusion by letting $\varepsilon \to 0$.

# F  Proof of Theorem 5

Denote $Y_{n+1} := T_{\nu_{n+1}}^{\mu_{n+1}}$ the optimal transport map between $\nu_{n+1}$ and $\mu_{n+1}$ and $\nabla_W\mathcal{H}(\mu_{n+1})$ the strong Fréchet subgradient of $\mathcal{H}$ evaluated at $\mu_{n+1}$ defined by Lemma 3: $T_{\mu_{n+1}}^{\nu_{n+1}} = I + \gamma\nabla_W\mathcal{H}(\mu_{n+1})$. Since $\mu_{n+1}, \nu_{n+1} \in \mathcal{P}_2^r(\mathcal{X})$, $(I + \gamma\nabla_W\mathcal{H}(\mu_{n+1})) \circ Y_{n+1} = I$ using Brenier's theorem. We thus have $\nu_{n+1}$-a.e.:

$$Y_{n+1} = I - \gamma\nabla_W\mathcal{H}(\mu_{n+1})(Y_{n+1}). \tag{29}$$

We firstly bound the $\mathcal{H}$ term. By taking $\mu = \mu_{n+1}$, $\pi = \mu_n$ and $\nu = \nu_{n+1}$ in Lemma 4, we have:

$$\mathcal{H}(\mu_{n+1}) \leq \mathcal{H}(\mu_n) - \langle\nabla_W\mathcal{H}(\mu_{n+1})(T_{\nu_{n+1}}^{\mu_{n+1}}), T_{\nu_{n+1}}^{\mu_n} - T_{\nu_{n+1}}^{\mu_{n+1}}\rangle_{\nu_{n+1}}. \tag{30}$$

We now identify $T_{\nu_{n+1}}^{\mu_n}$ and $T_{\nu_{n+1}}^{\mu_{n+1}}$. Recall that $Y_{n+1} = T_{\nu_{n+1}}^{\mu_{n+1}}$. Moreover, using Brenier's theorem and Lemma 2, $\nu_{n+1} \in \mathcal{P}_2^r(\mathcal{X})$ and $T_{\nu_{n+1}}^{\mu_n} = (I - \gamma\nabla F)^{-1}$. Therefore,

$$\mathcal{H}(\mu_{n+1}) \leq \mathcal{H}(\mu_n) - \langle\nabla_W\mathcal{H}(\mu_{n+1})(Y_{n+1}), (I - \gamma\nabla F)^{-1} - Y_{n+1}\rangle_{\nu_{n+1}}.$$

Using the transfer lemma, with $Y_{n+1} = X_{n+1} \circ (I - \gamma\nabla F)^{-1}$, the last inequality is equivalent to

$$\mathcal{H}(\mu_{n+1}) \leq \mathcal{H}(\mu_n) - \langle\nabla_W\mathcal{H}(\mu_{n+1})(X_{n+1}), I - X_{n+1}\rangle_{\mu_n}. \tag{31}$$

Then, we can bound the potential term. Using Equation (29), and $X_{n+1} = Y_{n+1} \circ (I - \gamma\nabla F)$, we have

$$X_{n+1} = I - \gamma\nabla F - \gamma\nabla_W\mathcal{H}(\mu_{n+1})(X_{n+1}). \tag{32}$$

Since $F$ is $L$-smooth, we have [4],

$$F(z) \leq F(x) + \langle\nabla F(x), z - x\rangle + \frac{L}{2}\|x - z\|^2, \qquad \forall\, x, z \in \mathcal{X}. \tag{33}$$

Replacing $z$ by $X_{n+1}(x)$, we obtain

$$F(X_{n+1}(x)) \leq F(x) - \gamma\langle\nabla F(x), \nabla F(x) + \nabla_W\mathcal{H}(\mu_{n+1})(X_{n+1}(x))\rangle \tag{34}$$

$$+ \frac{L\gamma^2}{2}\|\nabla F(x) + \nabla_W\mathcal{H}(\mu_{n+1})(X_{n+1}(x))\|^2. \tag{35}$$

Integrating w.r.t. $\mu_n$,

$$\mathcal{E}_F(\mu_{n+1}) \leq \mathcal{E}_F(\mu_n) - \gamma\langle\nabla F, \nabla F + \nabla_W\mathcal{H}(\mu_{n+1})(X_{n+1})\rangle_{\mu_n}$$

$$+ \frac{L\gamma^2}{2}\|\nabla F + \nabla_W\mathcal{H}(\mu_{n+1})(X_{n+1})\|_{\mu_n}^2. \tag{36}$$

Then, recalling (32) and summing equations (31) and (36), we get

$$
\begin{aligned}
\mathcal{H}(\mu_{n+1}) + \mathcal{E}_F(\mu_{n+1}) \leq & \mathcal{H}(\mu_n) + \mathcal{E}_F(\mu_n) \\
& - \gamma \langle \nabla_W \mathcal{H}(\mu_{n+1})(X_{n+1}), \nabla F + \nabla_W \mathcal{H}(\mu_{n+1})(X_{n+1}) \rangle_{\mu_n} \\
& - \gamma \langle \nabla F, \nabla F + \nabla_W \mathcal{H}(\mu_{n+1})(X_{n+1}) \rangle_{\mu_n} + \frac{L\gamma^2}{2} \| \nabla F + \nabla_W \mathcal{H}(\mu_{n+1})(X_{n+1}) \|_{\mu_n}^2 \\
\leq & \mathcal{H}(\mu_n) + \mathcal{E}_F(\mu_n) - \gamma \left( 1 - \frac{L\gamma}{2} \right) \| \nabla F + \nabla_W \mathcal{H}(\mu_{n+1})(X_{n+1}) \|_{\mu_n}^2 .
\end{aligned}
$$

## G  Proof of Lemma 6

Recall (29),

$$
Y_{n+1} = I - \gamma \nabla_W \mathcal{H}(\mu_{n+1})(Y_{n+1}) = T_{\nu_{n+1}}^{\mu_{n+1}}.
$$

Since $(Y_{n+1}, T_{\nu_{n+1}}^\pi)_\# \nu_{n+1}$ is a coupling between $\mu_{n+1}$ and $\pi$, we can upper bound the Wasserstein distance between $\mu_{n+1}$ and $\pi$ as:

$$
\begin{aligned}
W^2(\mu_{n+1}, \pi) \leq & \| Y_{n+1} - T_{\nu_{n+1}}^\pi \|_{\nu_{n+1}}^2 \\
= & \| I - T_{\nu_{n+1}}^\pi \|_{\nu_{n+1}}^2 - 2\gamma \langle \nabla_W \mathcal{H}(\mu_{n+1})(Y_{n+1}), I - T_{\nu_{n+1}}^\pi \rangle_{\nu_{n+1}} + \gamma^2 \| \nabla_W \mathcal{H}(\mu_{n+1})(Y_{n+1}) \|_{\nu_{n+1}}^2 \\
= & \| I - T_{\nu_{n+1}}^\pi \|_{\nu_{n+1}}^2 - 2\gamma \langle \nabla_W \mathcal{H}(\mu_{n+1})(Y_{n+1}), Y_{n+1} - T_{\nu_{n+1}}^\pi \rangle_{\nu_{n+1}} - \gamma^2 \| \nabla_W \mathcal{H}(\mu_{n+1})(Y_{n+1}) \|_{\nu_{n+1}}^2 .
\end{aligned}
\tag{37}
$$

where $\| I - T_{\nu_{n+1}}^\pi \|_{\nu_{n+1}}^2 = W^2(\nu_{n+1}, \pi)$. Moreover, using Lemma 4 with $\mu = \mu_{n+1}$ and $\nu = \nu_{n+1}$,

$$
-2\gamma \langle \nabla_W \mathcal{H}(\mu_{n+1})(Y_{n+1}), Y_{n+1} - T_{\nu_{n+1}}^\pi \rangle_{\nu_{n+1}} \leq -2\gamma \left( \mathcal{H}(\mu_{n+1}) - \mathcal{H}(\pi) \right).
$$

Plugging the latter inequality into (37), we get the result.

## H  Proof of Lemma 7

Since $(I - \gamma \nabla F, T_{\mu_n}^\pi)_\# \mu_n$ is a coupling between $\nu_{n+1}$ and $\pi$, we can upper bound the Wasserstein distance between $\nu_{n+1}$ and $\pi$ as:

$$
\begin{aligned}
W^2(\nu_{n+1}, \pi) \leq & \| (I - \gamma \nabla F) - T_{\mu_n}^\pi \|_{\mu_n}^2 \\
= & \| I - T_{\mu_n}^\pi \|_{\mu_n}^2 - 2\gamma \langle \nabla F, I - T_{\mu_n}^\pi \rangle_{\mu_n} + \gamma^2 \| \nabla F \|_{\mu_n}^2 .
\end{aligned}
\tag{38}
$$

where $\| I - T_{\mu_n}^\pi \|_{\mu_n}^2 = W^2(\mu_n, \pi)$. Moreover, since $F$ is $\lambda$-strongly convex, we have:

$$
F(x) \leq F(z) + \langle \nabla F(x), x - z \rangle - \frac{\lambda}{2} \| x - z \|^2, \qquad \forall \ x, z \in \mathcal{X}.
\tag{39}
$$

Replacing $z$ by $T_{\mu_n}^\pi(x)$ and multiplying by $2\gamma$, we obtain

$$
-2\gamma \langle \nabla F(x), x - T_{\mu_n}^\pi(x) \rangle \leq -2\gamma \left( F(x) - F \circ T_{\mu_n}^\pi(x) \right) - \gamma \lambda \| x - T_{\mu_n}^\pi(x) \|^2 .
$$

Integrating w.r.t. $\mu_n$ results in

$$
-2\gamma \langle \nabla F, I - T_{\mu_n}^\pi \rangle_{\mu_n} \leq -2\gamma \left( \mathcal{E}_F(\mu_n) - \mathcal{E}_F(\pi) \right) - \gamma \lambda W^2(\mu_n, \pi) .
$$

Plugging the latter inequality into (38) gives the result.

## I  Proof of Proposition 8

Recall that $Y_{n+1} = T_{\nu_{n+1}}^{\mu_{n+1}}$. Combining Lemma 6 and Lemma 7, we firstly get

$$
\begin{aligned}
W^2(\mu_{n+1}, \pi) \leq & (1 - \gamma \lambda) W^2(\mu_n, \pi) - 2\gamma \left( \mathcal{E}_F(\mu_n) + \mathcal{H}(\mu_{n+1}) - \mathcal{E}_F(\pi) - \mathcal{H}(\pi) \right) \\
& + \gamma^2 \| \nabla F \|_{\mu_n}^2 - \gamma^2 \| \nabla_W \mathcal{H}(\mu_{n+1})(Y_{n+1}) \|_{\nu_{n+1}}^2 .
\end{aligned}
\tag{40}
$$

Multiplying (36) by $2\gamma$,

$$-2\gamma\mathcal{E}_F(\mu_n) \leq -2\gamma\mathcal{E}_F(\mu_{n+1})$$
$$-2\gamma^2\langle\nabla F, \nabla_W\mathcal{H}(\mu_{n+1})(X_{n+1})\rangle_{\mu_n} - 2\gamma^2\|\nabla F\|_{\mu_n} + L\gamma^3\|\nabla F + \nabla_W\mathcal{H}(\mu_{n+1})(X_{n+1})\|^2_{\mu_n}.$$

Moreover, using the transfer lemma, $\|\nabla_W\mathcal{H}(\mu_{n+1})(X_{n+1})\|^2_{\mu_n} = \|\nabla_W\mathcal{H}(\mu_{n+1})(Y_{n+1})\|^2_{\nu_{n+1}}$. Therefore,

$$-2\gamma\mathcal{E}_F(\mu_n) + \gamma^2\|\nabla F\|^2_{\mu_n} - \gamma^2\|\nabla_W\mathcal{H}(\mu_{n+1})(Y_{n+1})\|^2_{\nu_{n+1}}$$
$$\leq -2\gamma\mathcal{E}_F(\mu_{n+1}) - \gamma^2\|\nabla F\|^2_{\mu_n} - \gamma^2\|\nabla_W\mathcal{H}(\mu_{n+1})(X_{n+1})\|^2_{\mu_n} - 2\gamma^2\langle\nabla F, \nabla_W\mathcal{H}(\mu_{n+1})(X_{n+1})\rangle_{\mu_n}$$
$$+ L\gamma^3\|\nabla F + \nabla_W\mathcal{H}(\mu_{n+1})(X_{n+1})\|^2_{\mu_n}$$
$$\leq -2\gamma\mathcal{E}_F(\mu_{n+1}) - \gamma^2(1 - L\gamma)\|\nabla F + \nabla_W\mathcal{H}(\mu_{n+1})(X_{n+1})\|^2_{\mu_n}.$$

Plugging the last inequality into (40),

$$W^2(\mu_{n+1}, \pi) \leq (1 - \gamma\lambda)W^2(\mu_n, \pi) - 2\gamma\left(\mathcal{E}_F(\mu_{n+1}) + \mathcal{H}(\mu_{n+1}) - \mathcal{E}_F(\pi) - \mathcal{H}(\pi)\right)$$
$$- \gamma^2(1 - \gamma L)\|\nabla F + \nabla_W\mathcal{H}(\mu_{n+1})(X_{n+1})\|^2_{\mu_n}.$$

## J   Proof of Corollary 10

Let $\mathcal{L}_n := 2\gamma n(\mathcal{G}(\mu_n) - \mathcal{G}(\mu_\star)) + W^2(\mu_n, \mu_\star)$. From Theorem 5, $\mathcal{G}(\mu_{n+1}) - \mathcal{G}(\mu_\star) \leq \mathcal{G}(\mu_n) - \mathcal{G}(\mu_\star)$ if $\gamma < 1/L$. Therefore,

$$2\gamma n(\mathcal{G}(\mu_{n+1}) - \mathcal{G}(\mu_\star)) + 2\gamma(\mathcal{G}(\mu_{n+1}) - \mathcal{G}(\mu_\star)) + W^2(\mu_{n+1}, \mu_\star) \leq 2\gamma n(\mathcal{G}(\mu_n) - \mathcal{G}(\mu_\star)) + W^2(\mu_n, \mu_\star),$$

where we used Theorem 9 with $\lambda = 0$ (recall that $\lambda \geq 0$). In other words, $\mathcal{L}_{n+1} \leq \mathcal{L}_n$. Finally,

$$2\gamma n(\mathcal{G}(\mu_n) - \mathcal{G}(\mu_\star)) \leq \mathcal{L}_n \leq \mathcal{L}_0 = W^2(\mu_0, \mu_\star).$$

## K   Proof of Corollary 11

Since the $\mathcal{G}(\mu_{n+1}) - \mathcal{G}(\mu_\star)$ is nonnegative, from Theorem 9,

$$W^2(\mu_{n+1}, \mu_\star) \leq (1 - \gamma\lambda)W^2(\mu_n, \mu_\star).$$

We get the result by iterating.