[Reviews · NeurIPS 2020]

Review 1

Summary and Contributions: This paper proposes a forward-backward discretization of Wasserstein gradient flow. === update === I gave an incorrect reference regarding the previous use of this method, the correct one should be: HANDLING CONGESTION IN CROWD MOTION MODELING http://cvgmt.sns.it/media/doc/paper/342/MRSV.pdf equations (18) (19). ====

Strengths: On the positive side, the paper is well written, studies an interesting problem, and the rates are what should be expected.

Weaknesses: On the negative side, I think there is almost no situation where the proposed scheme can be applied, which is probably the reason why this algorithm has never been really used and analyzed before. Also there are several existing works that use FW splitting.

Correctness: From a theoretical point of view, the paper is correct. The numerical evaluation only consider a very simple case (the only one possible?).

Clarity: Yes the paper is very well written (at least for a specialist on gradient flows).

Relation to Prior Work: === update === I gave an incorrect reference regarding the previous use of this method, the correct one should be: HANDLING CONGESTION IN CROWD MOTION MODELING http://cvgmt.sns.it/media/doc/paper/342/MRSV.pdf equations (18) (19). I have read the rebuttal, which helped to clarify some of the raised issues. I decided to keep the same score. ==== The use of Forward-backawrd discretization of JKO is not new. * It is used in [1,2,3]. These previous works do not assume that the smooth part is separable (i.e. the flow has a gradient structure). * It has been used in the context of Langevin sampling [6], the difference being that usually the forward step operates on the entropy part using a Brownian motion. This paper contains convergence speed analysis in the strongly convex case. See also [5] for a related work. *The litterature on proximal Langevin (for instance [7]), corresponding to a separable non-smooth function, is also relevant. [1] Splitting methods for systems of parabolic equations with nonlocal drifts in the nonpotential case, Laborde, Carlier, Nonlinear Analysis: Theory,Methods and Applications, 2017, [2] Weak Solutions to a Fractional Fokker-Planck Equation via Splitting and Wasserstein Gradient Flow, Malcolm Bowles, Martial Agueh [3] B. Maury, A. Roudneff-Chupin, F. Santambrogio, Congestion-driven dentritic growth, Discrete Contin. Dyn. Syst., Preprint 2014 [5] Andre Wibisono, Sampling as optimization in the space of measures: The Langevin dynamics as a composite optimization problem [6] Espen Bernton, Langevin Monte Carlo and JKO splitting [7] Marcelo Pereyra. Proximal Markov chain Monte Carlo algorithms. Statistics and Computing

Reproducibility: Yes

Additional Feedback: The main issue with this paper is that I think there is very few situations where the method can be applied. The only two that come to my mind are: - the function H(mu) is not diffusive, ie it is a separable function. Then the method boils down to independent foward-backward on the particles, which is well known and studied. - the function H(mu) is Ornstein-Uhlenbeck with a Gaussian initialization, which is basically the setup considered by Andre Wibisono in "Sampling as optimization in the space of measures", which is also what is done in the numerical part of this paper. So I think the paper should either show other potential non trivial uses of their algorithm, and maybe oversell a bit less their contributions. In its current form, it does not really explain why is could be interesting for the ML community. Other remarks: - What about extension to non-separable but smooth (in the Wasserstein sense) convex functions ? [1,2,3] do not assume this and consider flows which are not of gradient type. - "Broader impact" should not be a conclusion.


Review 2

Summary and Contributions: This paper studies the problem of minimizing a composite functional G = E + H over the space of probability measures with a finite second moment, where E is a linear functional defined by some function F (called the potential) and H is convex but possibly non-smooth functional of measures. The paper proposes to solve the problem by the (discrete-time) forward-backward algorithm which is constructed using the JKO operator. The major contribution of the paper is the non-asymptotic analysis of the forward-backward algorithm for this problem. In particular, the paper shows that, under a set of assumptions, the algorithm converges sublinearly and linearly if the potential function F is convex and strongly convex, respectively.

Strengths: As far as I know, the main theoretical result (the convergence rate of the forward-backward algorithm for the considered minimization problem) is new. One noteworthy aspect about the main results is that, unlike many other approaches for solving optimization problems over the Wasserstein space whose theoretical guarantees are obtained based on continuous dynamics, the paper obtained non-asymptotic convergence rates for the discrete-time algorithm.

Weaknesses: Although the paper advocate the idea that the JKO operator can be computed and implemented efficiently, this is not the case, to the best of my knowledge. In fact, the only efficiently computable JKO operator that I am aware of is the relative entropy with respect to a Gaussian distribution, as mentioned in Remark 1. This largely limits the practicality of the proposed approach, which is also manifested in the rather limited experimental results. If there are other examples of H whose JKO operator are easy to compute, the paper should present them. A weakness in terms of the theory is that the discussions and theoretical results about the non-smooth convex functional H and its JKO operator is not enough, adding which would be beneficial to both the theorists and practitioners in the community. Two suggestions to address this are as follows. First, the paper could provide some concrete examples (with formulas written down, possibly in Appendix if not enough space) of interaction energies and regularizers that arise from machine learning applications. Second, it would be good to discuss more on the assumptions B2 and B3 (for which functional H they hold? are there easily verifiable sufficient conditions?). UPDATE: ------------------------------------ It's a bit unfair to compare the JKO operator with the proximal operator and hence justify the paper by saying that it motivates the studies of JKO operator. After all, the intrinsic nature and complexity for solving the proximal operator (a finite-dimensional problem) and JKO operator (an infinite-dimensional problem) are very different. Also, I'm still not satisfied by the neural network example provided in the response. As pointed out, it is not covered by the theory in the paper.

Correctness: The theoretical results are correct as far as I could verify. As for the numerical experiments, it is hard to judge as much details are missing. For example, the computer environment for conducting the experiments is not given. See also the comments below regarding the numerical section.

Clarity: The paper is generally well written. However, some parts might be difficult for non-experts (of optimal transport) to follow. For example, the concept of convexity along generalized geodesics is not very easy to understand as it is not a direct generalized of the usual Euclidean convexity. Also, some details for the numerical section are missing, such as the computer environment for the experiments, the procedure of particle implementation and the relation among \mu_n, the particles and the parameters (m_n, \sigma_n).

Relation to Prior Work: It would be good to clarify the relation between the theoretical result of the paper and that of references [11] and that in the Appendix G of reference [34], which both studied discrete-time algorithms for some optimization problems on the space of measures.

Reproducibility: No

Additional Feedback: The references contain many typo/errors. Many of the Wasserstein's, Langevin's, Fokker's, Planck's, etc., are not capitalized. Many of the book or journal titles are also not properly capitalized. For reference [34], it is better to use the COLT and the arXiv.


Review 3

Summary and Contributions: This paper generalizes the Forward Backward (FB) algorithm from the Euclidean space to the Wasserstein one. A long introduction of the problem is given in Section 2. The algorithm is presented in Section 3 and the main results are summarized in Section 4. Final experiments illustrate numerically the convergence on a toy example as well as the convergence rate presented in the final corollary 11.

Strengths: I find this work original and elegant. The extension of the FB algorithm to the Wasserstein space is new. With respect to the Euclidean case, the proofs rely on different tools (use of transport map instead of lipschizianity of the F and B operators for the Euclidean case).

Weaknesses: The section 2.3 is long. It makes an interesting parallel between Euclidean and Wasserstein cases but it could be a bit reduced (for instance the convexity property (5) ...). In Section 4, it would be nice to follow this parallel and to add a short discussion on why the approach for proving the convergence in the Euclidean space can not be naturally generalized to the Wasserstein space. It would be interesting to add a few other concrete example of problems where the proposed scheme can be useful. In section 5 (or in the appendix), it is necessary to add more details on the implementation.

Correctness: All the proofs seem correct.

Clarity: The paper is very well written. The section 2 gives a general presentation of the problem accessible for non specialists.

Relation to Prior Work: The relations with previous works seem adequate, but I am not a specialist of the recent literature.

Reproducibility: Yes

Additional Feedback: Line 53: JKO should be detailed here and not 73. Line 59: SVGD is not defined. Line 157 (e before the equation): EVI has already been defined Line 226, it should be written that X_{k+1}= T_k^{k+1}


Review 4

Summary and Contributions: The authors present a Forward-Backward discretization scheme for gradient flows between probability measures in Wasserstein space, based on the forward-backward Euler scheme in Hilbert Space. They analyze the properties of that discretization in section 4 and prove that the rates of convergence for that scheme are similar to those in Hilbert space.

Strengths: Mathematical rigor - the paper is rigorous in its presentation and meticulously proves all claims. The empirical experiments are not particularly convincing but the contributions are theoretical, with very little dependence on empirical methodology. 2 Citations are excellent. I had to refer to the related work several times and in each case the authors pointed to a perfectly relevant passage. 3 Advances in Optimal Transport are frequently published in NeurIPS. The subject matter is clearly relevant to the community.

Weaknesses: The contributions unique to this paper appear to be mostly incremental. In the introduction, the authors describe their algorithm as a generalization of one previously presented and that their main contribution is to prove non-asymptotic convergence rates for that algorithm, doing so in section 4.4, in which they find that covergence in Wasserstein space is similar to that of proximal gradient methods in Hilbert space. 2 The experiments are mostly trivial and are barely analyzed. The authors provide the code to generate all experiments and visualizations in their supplementary material (and it works!), but as in the paper, there's minimal documentation supporting the experiments. As it exists now, there's very little added value in the experiments section. 3 There appears to be very limited direct application for this work in practice, due to the massive computational complexity of these measures.

Correctness: The claims and methodology appear to be correct, and make sense from a high-level perspective: the authors are presenting a forward-backward Euler scheme in Wasserstein space, using appropriate operators that correspond to the normal Euler operators (the pushforward operator for the gradient step and the JKO operator for the proximal step). But this paper is out of my area; if there were mistakes in the proofs it is unlikely that I would identify them. The assumptions do seem to be mild, as the authors claim. It would be nice to have a few lines to establish some intuition with respect to how reasonable the assumption that the regularizer H is convex along a generalized geodesic in Wasserstein space is. It appears that this H is almost always the negative entropy in practice? The broader impact statement suggests that more research into forms for the H functional is necessary.

Clarity: The paper and appendix are clearly expertly written. The presentation is dense, making it difficult to access for readers who are not already experts in convex optimization and optimal transport - this includes the sections meant to review those topics and establish preliminaries. The authors may have struggled to get the content into the page limit.

Relation to Prior Work: The authors do a nice job here. Even though I am not in the area I feel confident that I understand which ideas are new to this paper and which came from other areas.

Reproducibility: Yes

Additional Feedback: As stated above, this paper is out of my area and I am not able to verify the correctness of the proofs definitively. The strengths and weaknesses above may seem contradictory to my scoring of the paper as marginally above the acceptance threshold, but I think that despite the fact that there isn't much novelty here and the experiments are weak, there is value in the inclusion of purely theoretical papers that are obviously well-researched like this one, even if the takeaway from the paper is mostly lines 242-246. I'm putting my score in the middle of the range because of my low confidence. Some suggestions: as written, this paper sits in a bit of an awkward postition on the spectrum between accessibility and technical depth, with respect to what the authors require from a reader. I think the reviews of Optimal Transport and Gradient Flows aren't necessary for inclusion at this level of technicality - a reader familiar with those topics won't need this review, and a reader unfamiliar with the topics won't get much out of the formal presentation and will have to look to other sources anyway. The paper would probably be more accessible, not less, if the authors recoup some of this "review" space to add more to sections 3 and 4 and highlight their own contributions. The experiments section really doesn't add anything to the paper. It feels like it's there because the authors felt it necessary to have such a section because they're submitting to NeurIPS, but the theoretical results stand on their own; this paper would probably be better with the experiments section deleted and that space being used in another way. There's a bit of a tradeoff with section 2 available here: section 2 could be improved by the removal of 5, or 5 could be improved by the reworking of 2. The latter option would take the paper in a direction for more empirical evaluation of the setting.

[Author Response · NeurIPS 2020]

We thank Reviewers (R) 1, 2, 3, and 4 (who gave us marks 7, 6, 8, 6 respectively) for their positive feedback on the
quality and clarity of the paper and their pertinent suggestions. About its writing, we are taking your remarks into
account to improve the structure of some sections and we will correct the writing defaults. We will also add details
on the concepts (e.g. R2-generalized geodesic convexity, and R3-why the proof differs from the Euclidean case) and
discussion on the practical implementation. We first want to recall that our main contribution is the analysis of the FB
scheme as an optimization algorithm to minimize $\mathcal{G} = \mathcal{E}_F + \mathcal{H}$. For instance, if $\mathcal{H}$ is the negative entropy, FB is: (i)
faster than Langevin Monte Carlo (LMC), although we will acknowledge that it has an higher iteration complexity; (ii)
unbiased (i.e. converges to $\mu^\star$ with a constant step) unlike LMC, because the F and B operators are adjoint to each
other (see [34, Sec. F.1]). As R2 says, our work contrasts with many contributions on optimization over the Wasserstein
space that only obtain guarantees in continuous time, which corresponds to an idealized setting.

**R1, R2, R3, R4. Examples of functionals $\mathcal{H}$ in ML (other than entropies).** We shall give formulas for the negative
entropy and higher order entropies $\mathcal{H}$ that are natural regularizers. Besides, consider a Infinite-width 1 hidden layer
neural network (1HLNN, see [3,12,23]) and let $n$ be the number of neurons. For any input $x$, the output of the 1HLNN
can be written as $f(x) = \frac{1}{n}\sum_{i=1}^n \phi(x, z_i)$, where $\phi(x, z_i) = w_i \psi(x, \theta_i)$ with $w_i \in \mathbb{R}$ the weight of the $i$-th neuron and
$\theta_i \in \mathbb{R}^d$ parametrizes the activation function $\psi$ (e.g. a sigmoid). Given (regression) data $(x, y) \sim p$, the optimization
of this NN can be written as the minimization of the MSE: $\text{argmin}_{z_1,\ldots,z_n \in \mathcal{Z}} \frac{1}{2}\mathbb{E}_{(x,y)\sim p}[\|y - \frac{1}{n}\sum_{i=1}^n \phi(x, z_i)\|^2]$,
where $\mathcal{Z} = \mathbb{R} \times \mathbb{R}^d$. When $n \to \infty$, this becomes an optimization problem over $\mathcal{P}(\mathcal{Z})$ (the set of probability
measures over $\mathcal{Z}$) : $\text{argmin}_{\mu \in \mathcal{P}(\mathcal{Z})} \frac{1}{2}\mathbb{E}_{(x,y)\sim p}[\|y - \int \phi(x, z)d\mu(z)\|^2]$. Expanding this loss function leads to the
objective $\mathcal{G}(\mu) = \int F(z)d\mu(z) + \frac{1}{2}\int K(z, z')d\mu(z)d\mu(z')$, where $F(z) = -\mathbb{E}_{(x,y)\sim p}[y\phi(x, z)]$ is a potential and
$K(z, z') = \mathbb{E}_{x \sim p_x}[\phi(x, z)\phi(x, z')]$ is an interaction term. Note however that this example is not strictly covered by our
theory as $\mathcal{G}$ is only $\lambda$-geodesically convex (with $\lambda < 0$, see [12]).

**R2, R4. Computation of JKOs.** We agree and will acknowledge that the cases given by R1 and R2, as far as we know,
are the only ones where exact formulas of JKOs are known. However, there is an extensive literature on the computation
of JKOs of *generic* functionals using subroutines, see e.g. the review [29, Section 4.8] and the more recent work [27].
Just as proximal methods in Euclidean optimization, the FB scheme relies on subroutines to compute the JKO step. Our
results on FB scheme could motivate research on specific JKOs relevant to ML. **Assumptions B1–B3.** B1-B2 are just
general technical assumptions used in [2] (see [2, Eq 10.1.1a, Eq 10.1.1b]) that are always satisfied in relevant cases.
Then [2, Prop 9.3.2, 9.3.5 and 9.3.9] gives broad examples of (potential, interaction and internal) energies satisfying B3,
e.g., potential (resp. interaction) energies if the potential (resp. interaction) term is convex, and entropies.

**R2, R3, R4. Experiments.** We only provided simulations in a toy model, in low and high-dimensions, to illustrate
some of our results such as the predicted linear convergence (Fig. 2). Further numerical investigations would imply
integrating subroutines to tackle problems with more complicated JKOs, and will be the subject of future work. We will
add documentation about the experiments, including the computer environment and the update formulas obtained from
[34, App. G], and discuss more precisely our numerical results.

**R1. Related work.** In the following, [[]] denotes references mentioned in the reviews while the ones of the paper are
still denoted by []. These splitting methods are indeed related, we will cite the missing references [[1,2,3,6]]. However,
we stress that to the best of our knowledge, the FB scheme is new, except in the specific case where $\mathcal{H}$ is the negative
entropy [34, App. G]. Also, it is not covered in [[1,2,3,6]]. The method of [[1]] employs the JKO of $\mathcal{E}_F + \mathcal{H}$ instead of
splitting $\mathcal{E}_F$ and $\mathcal{H}$. Then, [[2]] and variants of the Langevin algorithm [[4,5,6]] are not FB because they use a flow
step for $\mathcal{H}$, i.e. a step where a distribution is transported to another one following the *exact* gradient flow of $\mathcal{H}$. Indeed,
in the Langevin algorithm, $\mathcal{H}$ is the negative entropy and the exact gradient flow is given by the Brownian motion
(see [34, Sec. 2.2.2 vs Sec. 4.1] for the difference between FB and Langevin). Finally, [[3]] proposes an alternating
scheme (Eq. (10) therein) which also differs from the FB scheme. **Non separable $\mathcal{E}_F$.** This is an interesting question.
If $\mathcal{E}_F$ is replaced by some $\mathcal{F}$ not separable, i.e. $\mathcal{F}(\mu) \neq \int F(x)d\mu(x)$, then $\nabla\mathcal{F}(\mu) \neq \nabla F$ and Eq. (17) becomes
$\nu_{n+1} = (I - \gamma\nabla\mathcal{F}(\mu_n))_\#\mu_n$. Extending our results is straightforward if $\mathcal{F}$ is convex along any interpolating curve (e.g.
interaction energies with convex term [2, Prop. 9.3.5]) because one can still upper bound scalar products between $\nabla\mathcal{F}$
and *non-optimal* pushforwards (arising in the FB scheme). If $\mathcal{F}$ is only convex along geodesics, this is more difficult.

**R2. Related work.** We shall provide more details on [11,34]. [11] considers the resolution of the Wasserstein
barycenter problem, which corresponds to a non geodesically convex objective, via gradient descent. This is done using
nonconvex optimization techniques, but not using JKO steps. [34, App. G] inspired us the FB scheme. We extended it
to a general $\mathcal{H}$ and established convergence rates (which were unknown even in the particular case of [34, App. G]).

**R4. Contribution.** We appreciate your positive comments on the inclusion of such papers to NeurIPS. We think
that our contribution is not as incremental as it may look. The FB scheme was proposed in a particular case but was
never analyzed as an optimization algorithm (even in the particular case, [34, App. G]). It is also surprising that the
convergence rates are similar to the Hilbert case: LMC, which deals with $\mathcal{H}$ being the negative entropy, does not have
this property. The proof requires non trivial ideas that are interesting on their own.

[Meta-Review · NeurIPS 2020]

The reviewers all found the theoretical contribution to be of value to the NeurIPS community and recommended an accept. Note that there are several important comments from the reviewers that should be taken into account in the final version.